# Efficacy and safety of temperature-sensitive acellular dermal matrix in prevention of postoperative adhesion after thyroidectomy: A randomized, multicenter, double-blind, non-inferiority study

Jin Kyong Kim[1], Cho Rok Lee[1¤a], Sang-Wook Kang[1], Jong Ju Jeong[1], Kee-Hyun Nam[1]*, Sung-Rae Cho[2], Seongmoon Jo[2], Eun Young Kim[3], Ji-Sup Yun[3], Hee Jin Park[4], Mi Sung Kim[4], Kwangsoon Kim[5], Sohee Lee[5¤b], Ja Seong Bae[5], So Yeon Jun[6], Jihye Park[6¤c], Jeong Soo Kim[5]

1 Department of Surgery, Severance Hospital, Yonsei Cancer Center, Yonsei University College of Medicine, Seoul, Korea, 2 Department and Research Institute of Rehabilitation Medicine, Severance Hospital, Yonsei University College of Medicine, Seoul, Korea, 3 Department of Surgery, Kangbuk Samsung Hospital, Sungkyunkwan University School of Medicine, Seoul, Korea, 4 Department of Radiology, Kangbuk Samsung Hospital, Sungkyunkwan University School of Medicine, Seoul, Korea, 5 Department of Surgery, Seoul St. Mary's Hospital, College of Medicine, The Catholic University of Korea, Seoul, Korea, 6 Department of Rehabilitation Medicine, Seoul St. Mary's Hospital, College of Medicine, The Catholic University of Korea, Seoul, Korea

¤a Current address: Department of Surgery, Yongin Severance Hospital, Gyeonggi-do, Korea
¤b Current address: Department of Surgery, Eunpyeong St. Mary's Hospital, College of Medicine, The Catholic University of Korea, Seoul, Republic of Korea
¤c Current address: Department of Rehabilitation Medicine, Eunpyeong St. Mary's Hospital, College of Medicine, The Catholic University of Korea, Seoul, Republic of Korea
* KHNAM@yuhs.ac

## Abstract

### Introduction

MegaShield® is a newly developed temperature-sensitive anti-adhesive containing micronized acellular dermal matrix. The aim of this study was to investigate the efficacy and safety of MegaShield® compared with Guardix-SG® in the prevention of adhesions in patients undergoing bilateral total thyroidectomy.

### Method

We conducted a multicenter trial between October 2018 and March 2020 in patients undergoing total thyroidectomy. The patients were randomly assigned to either the MegaShield® group or the Guardix-SG® group. The primary outcome was the esophageal movement using marshmallow six weeks after the surgery and the secondary outcome was the assessed adhesion score. The safety assessment was also evaluated.

### Results

The study included 70 patients each in the MegaShield® and control (Guardix-SG®) groups. Baseline clinical characteristics, the mean score of marshmallow esophagography, and the

**Data Availability Statement:** All relevant data are within the article and its Supporting Information files.

**Funding:** KHN This research was supported by a grant of the Korea Health Technology R&D Project through the Korea Health Industry Development Institute (KHIDI), funded by the Ministry of Health & Welfare, Republic of Korea (grant number: HI18C2359) The funders had no role in study design, data collection and analysis, decision to publish, or preparation of the manuscript.

**Competing interests:** The authors of this manuscript have read the journal's policy and declare the following competing interests: this study's protocol and test devices were sponsored by L&C bio; however, a grant was finally sponsored by the Ministry of Health & Welfare. There are no patents, products in development, or marketed products to declare. This does not alter our adherence to PLOS ONE policies on sharing data and materials.

sum of adhesion scores were not statistically different between the two groups. Inferiority test demonstrated that the efficacy of MegaShield® is not inferior to that of Guardix-SG®. There were no device-related complications in both groups.

## Conclusion

The efficacy and safety of MegaShield® were not inferior than those of Guardix-SG®. Mega-Shield® demonstrated the potential of ADM as a potential future anti-adhesive agent.

## Trial registration

The name of trial registry CRIS (Clinical Research Information Service) https://cris.nih.go.kr/cris/index.jsp. (The full trial protocol can be accessed) Registration number: KCT0003204.

## Introduction

Postoperative adhesion refers to the formation of fibrous bands between the traumatized tissues after surgery. It is a natural phenomenon due to the proliferation and regeneration of the tissue cells which occurs after a surgical procedure. The severity of postoperative adhesions can vary, resulting in numerous complications. Sometimes, it can be hazardous and even life-threatening, leading to malfunction of organs which require re-intervention.

Surgical intervention for thyroid gland is inevitable when a patient has uncontrolled hyperthyroidism or thyroid cancer, the most common endocrine cancer worldwide [1]. Although postoperative adhesions after thyroid surgery are not life-threatening, many patients experience life-long discomfort such as pain, stiffness, hyperesthesia, paresthesia, swallowing difficulty, and in some severe cases, choking sensation. Attempts to prevent postoperative adhesion formation and associated complications include minimizing the dissection during surgery; administering anti-inflammatory agents, fibrinolytic agents, anticoagulants, and other hormonal drugs; and applying anti-adhesive agents around the surgical field [2–8].

Although complete recovery of postoperative adhesion is not possible, various anti-adhesion barriers (AABs) that are currently available are known to be helpful in reducing the extent of postoperative adhesion. AABs are divided into three categories according to their viscosity: film/membrane, solution, and gel [9–11]. Each AAB presents different disadvantages: film/membrane type shows foreign body reaction, application difficulties, and adhesion formation at the suture site; solution type may cause inaccurate application; and the gel type shows short maintenance time due to rapid dissolvement and discharge [12–14]. Nowadays, temperature-sensitive anti-adhesives using poloxamer [15], which has sol-gel transition at body temperature, has been developed, of which Guardix-SG® is the most common example.

Similarly, acellular dermal matrix (ADM) has been studied for its usefulness as an anti-adhesive agent [16–18]. ADM is derived from donated human skin supplied by the US tissue banks following the guidelines of the American Association. ADM contains various substances such as collagen and elastin, and acts as a biologic platform for re-epithelialization, neovascularization, and fibroblast infiltration while not inducing an immune response [16, 19, 20]. ADM has been applied in various surgical fields such as abdominal wall repair [21], dural repair [22], pelvic floor reconstruction [23], and bladder wall repair [24] for minimizing postoperative adhesion and facilitating wound healing and soft tissue augmentation.

In this context, we developed a new anti-adhesive agent which includes micronized ADM with sol-gel transition properties for effective application. The resulting product is Mega-Shield® (L&C BIO, Seongnam, Korea), consisting of ADM, poloxamer 407, sodium hyaluronate (HA), and 1,4-butanediol diglycidyl ether (1,4-BDDE). Our previous study demonstrated that MegaShield® is effective as an anti-adhesive in a rat model [17]. In this study, we demonstrate the efficacy and safety of MegaShield® in a human model. We hypothesized that the efficacy and safety of MegaShield® are not inferior to those of Guardix-SG®, and the aim of this study was to investigate our hypothesis using a randomized, multicenter, double-blind, non-inferiority study design.

## Materials and methods

### Patients

Between October 4, 2018 and March 26, 2020, patients who underwent total thyroidectomy were enrolled and followed up in this study. This multicenter trial was conducted by three surgeons in three different hospitals. Patients with the following were excluded: < 20 years old or ≥ 70 years old, liver or kidney function abnormality, previous history of lymphatic or hematologic disease, diabetes mellitus, immune system suppression, autoimmune disorder, severe systemic disorder, anticoagulative medication intake, chemotherapy for other cancer, or expectation of other following surgery. The study protocol was approved by each Institutional Review Board of Severance hospital (IRB Approval No. 1-2018-0034), Kangbuk Samsung Hospital (IRB Approval No. KBSMC 2018-05-028), and Seoul St. Mary's Hospital (IRB Approval No. KC18DDDT0299), Korea. No changes to methods after trial commencement are implemented.

### Products used in this study

Both MegaShield® and Guardix-SG® are temperature-sensitive anti-adhesive agents which can transform from solution to gel form at body temperature. MegaShield Ⓡ comprises ADM, poloxamer 407, HA, and 1,4-BDDE. Guardix-SG Ⓡ does not contain ADM and comprises poloxamer, sodium alginate, and calcium chloride. The patients received 5 g (5 mL) of Mega-Shield® or Guardix-SG® according to their assigned group.

### Randomization and blinding

The patients were randomly assigned to either the MegaShield® group or the Guardix-SG® group after the anesthetic effect subsided, and the randomization was performed at a 1:1 ratio through a permuted block randomization method with a fixed block size of four. The randomization was stratified by center and performed with SAS ver. 9.4 (SAS Institute, NC, USA) software by a professional statistician of the Clinical Trials Center Yonsei University Health System. Patient identification codes were assigned only to patients who were judged to meet the inclusion/exclusion criteria among those who needed total thyroidectomy. In addition, patients were given a randomization code (3 digits) in order according to the randomization table and assigned to the corresponding application group. The randomization code and patient identification code were used as the patient number during the clinical trial. During the study, the surgeons who performed the surgeries were not blinded as the packaging of the two devices were different. Meanwhile, patients and evaluator were blinded. The clinical assessor was an independent evaluator who did not participate in the surgery; therefore, it was not possible for him/her to know which investigational product had been applied to the patient. Since he/she contacted the patient only during evaluation, blinding was maintained. Since the

assigned investigational product was applied in the patient while under general anesthesia during surgery, he/she could not know which investigational product had been applied, even after the surgery.

## Surgical procedure

Standard bilateral total thyroidectomy was performed in all patients. Patients underwent open thyroidectomy with central lymph node dissection by a single surgeon (K.-H.N. in Yonsei college of Medicine, J.S.B in Seoul St. Mary's Hospital, and J.S.Y in Kangbuk Samsung Hospital) in each of the centers. After total thyroidectomy, 2 mL of MegaShield® or Guardix-SG® was applied between the space of tracheal wall and strap muscle according to their group. After closing the midline of the strap muscles, 3 mL of the identical antiadhesive agent was applied between the overlying subcutaneous fat layer and strap muscle fascia.

## Data collection and interventions

All data were prospectively collected by each of the three clinical assessors in each institution. Clinical visits were conducted for 30 days at 0 day before the surgery (visit 1), day of the surgery (visit 2), postoperative week 1 (visit 3), and week 6 (visit 4). During visit 1, the baseline demographic characteristics and clinical and laboratory assessments were thoroughly recorded after obtaining written informed consent. During visit 2, the investigational device (Mega-Shield® or Guardix-SG®) was applied on the surgical field. Participants were followed up twice (visits 3 and 4) after device application. Clinical and laboratory variables and adverse events were evaluated during all visits.

## Outcome measures

Patient demographic findings including age, sex, and body mass index were documented. The primary outcome was the esophageal movement using marshmallow [25] at visit 4. With this evaluation, the affected esophageal motility due to postoperative adhesion was checked. Esophageal motility was classified as normal and scored 3 if the esophageal transit time of the bolus marshmallow in the prone position occurred within 30 seconds. The specified score classification of the marshmallow esophagography is presented in Table 1. The mean scores of both groups were compared. The secondary outcome was the assessed adhesion score at visits 1, 3, and 4, presenting the questionnaire survey to both the patients and the clinical assessors. The questionnaire included subjective discomforts in swallowing, cosmetic changes in cervical skin wrinkles answered by the patients, and objective adhesion severity evaluated by the clinical assessors. The evaluation criteria are presented in Table 2. Each criterion was assessed using a score from 0 to 10, in which a higher score represented a more severe adhesion. The sum of all scores and the change in scores from the screening day (visit 1) after surgery were compared between the two groups. The safety outcomes included adverse events and changes in laboratory findings, vital signs, and physical examination. All data on these outcomes were collected

**Table 1. Classification of marshmallow esophagography.**

| Score | Definition | Transition time of marshmallow |
|---|---|---|
| 3 | Normal | Transition within 30 seconds in supine position |
| 2 | Mild | Transition after 30 seconds in supine position and within 30 seconds in erect position |
| 1 | Moderate | Cessation in supine position and transition after 30 seconds in erect position |
| 0 | Severe | Cessation in both supine and erect positions |

**Table 2. Adhesion score questionnaire.**

| Evaluation criteria | |
| --- | --- |
| Subjective discomforts of patients | 1. Is there difficulty in swallowing saliva? |
| | 2. Is there difficulty in swallowing water? |
| | 3. Is there difficulty in swallowing solid food? |
| | 4. Do wrinkles around cervical area look abnormal? |
| Objective evaluation of clinical assessor | 5. Do wrinkles around cervical area look symmetrical and natural with natural position? |
| | 6. Do wrinkles around cervical area look symmetrical and natural with neck extension? |
| | 7. Is there inflammatory reaction or scar formation? |

by the clinical assessors. No changes to trial outcomes after trial commencement are implemented.

## Statistical analysis

The sample size was calculated to perform the non-inferiority test [26]. We supposed that the esophageal movement score (marshmallow esophagography score) of the test group (Mega-Shield) was not lower than that of the control group (Guardix-SG$^®$) after thyroidectomy. Group sample sizes of 63 each achieved 80% power to detect non-inferiority if the lower limit of the one-sided 97.5% confidence interval (CI) of the mean difference was less than -0.1 (non-inferiority margin) using two-sample unpaired t-test. The non-inferiority margin was calculated based on a previous study [27]. In the study, the difference in scores between comparator (Guardix-SG$^®$) and placebo was 2.93–2.73 = 0.2; therefore, assuming that, the margin is 0.2/2 = 0.1. Hence, we assumed the standard deviation of 0.2, non-inferiority margin of -0.1, and equal mean difference between the two groups. After accounting for an overall dropout rate of 10%, a total of 140 patients were required. The analysis sets consisted of intent-to-treat (ITT), modified intent-to-treat (mITT), and per protocol (PP). The ITT set included any patient who was treated at least once in the assigned group, while the mITT set consisted of any patient who belonged to the ITT set and was followed up for the first evaluation for estimating the effectiveness of the medical device of the assigned group. The PP set included any patient who belonged to the ITT and mITT set and completed the trial without any major protocol violations. The efficacy analysis was performed primarily on the mITT population, and concomitantly on the PP population. A safety analysis was performed on the ITT set. For primary outcome, 97.5% CI was established with unpaired t-test. For secondary outcome and safety analysis, chi-squared test or Fisher's exact test was applied for categorical data, and unpaired t-test or Wilcoxon signed rank test was used depending on whether the normality assumption was satisfied (Shapiro-Wilk test) for continuous data. The significance level of each statistical analysis was set at $p<0.05$. SAS ver. 9.4 was used for statistical computation.

## Results

### Patient characteristics

The patients were enrolled between October 4, 2018 and February 11, 2020, and followed up until March 26, 2020. The patient enrollment is depicted in the flow diagram as shown in Fig 1. Among the 156 recruited patients, 16 were excluded due to screening failure and 140 were randomly assigned to either the MegaShield$^®$ group (n = 70) or the Guardix-SG$^®$ group (n = 70) and analyzed as the ITT set. There were 70 patients from Severance hospital, 44 from

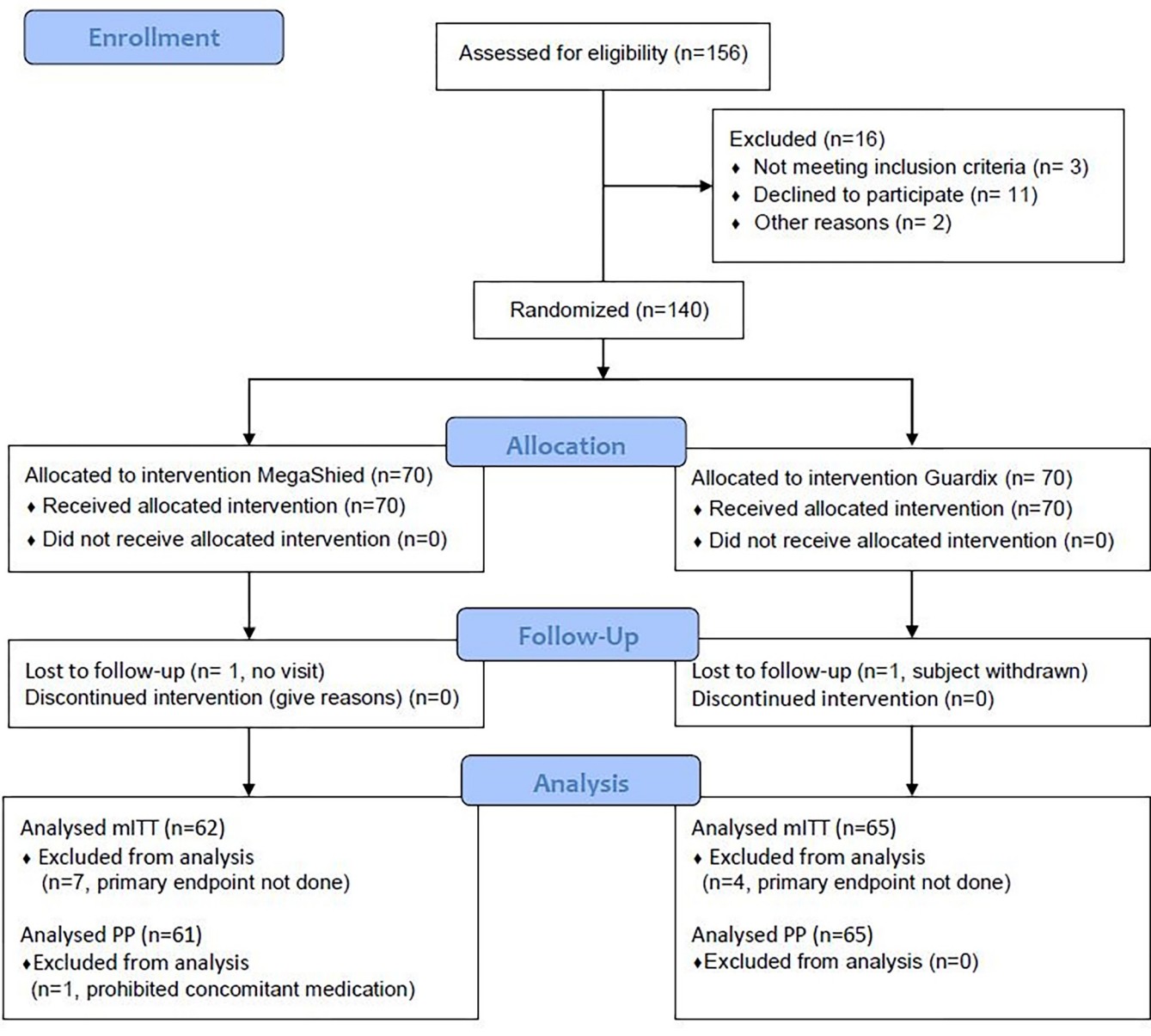

**Fig 1. Disposition of patients.**

Kangbuk Samsung hospital, and 26 from Seoul St. Mary's Hospital (S1 Table). No deviation occurred in the randomization process. One patient in the MegaShield® group was excluded due to loss of follow-up. One patient in the Guardix-SG® group was excluded due to the patient's request after surgery. During the first efficacy evaluation, the marshmallow esophagography, 11 patients (7 patients in MegaShield® group and 4 in Guardix-SG® group) who swallowed saliva multiple times within 30 seconds or did not swallow saliva even after 30 seconds were considered as protocol deviation. Thus, mITT analysis was conducted on 127 patients (62 in MegaShield® group and 65 in Guardix-SG® group). After marshmallow esophagography, one patient in MegaShield® group was found to have consumed aspirin and was excluded from the study. Hence, PP set included 126 patients (61 in MegaShield® group and 65 in

Guardix-SG® group). There were no significant differences in baseline clinical characteristics between the two groups (Table 3).

## Efficacy results

As the primary efficacy outcome, the outcomes of the marshmallow esophagography were compared at visit 4 (Table 4). The mean score was 2.8 ± 0.6 in MegaShield® group and 2.6 ± 0.8 in Guardix-SG® group in mITT (p = 0.16) and PP set (p = 0.17) analyses. The 97.5% one-sided CI of the difference was -0.08–Inf., and the calculated lower limit of the one-sided CI exceeded the prespecified noninferiority margin of -0.1 demonstrating the noninferiority of MegaShield® on mITT set. Similarly, the 97.5% one-sided CI of the difference was -0.07–Inf., and demonstrated the noninferiority of MegaShield® on PP set.

The secondary outcome, adhesion scores, was assessed at visits 1, 3, and 4 using questionnaire survey in both the patients and the clinical assessors. Shapiro-wilk test (p<0.001) and Wilcoxon test were applied because the sum of the adhesion scores were not normally distributed. In mITT and PP sets, there were no significant differences between the sum of adhesion scores in every visit between the MegaShield® group and the Guardix-SG® group (Table 5).

## Safety results

We compared the incidence of postoperative complications and adverse events with the ITT set analysis. Table 6 shows the summary of the adverse events. There were no significant differences in rates of any adverse event or serious adverse events between the two groups. Although adverse events occurred in 35 (50.0%) patients with 82 cases in MegaShield® group and 38 (54.3%) patients with 91 cases in Guardix-SG® group, most of the cases belonged to mild adverse events. Moderate adverse events occurred in one patient in MegaShield® group who had aspartate aminotransferase (AST) and alanine aminotransferase (ALT) elevation counted as two adverse event cases, in which the elevated levels recovered. In Guardix-SG® group, moderate adverse event occurred in one patient who had postoperative shoulder pain. Four serious adverse events, which resulted in prolonged hospital stay due to the exacerbation of the general condition of the patients, occurred in two patients of the MegaShield® group. One patient had postoperative fever (38.1–38.4˚C) and the other patient had postoperative fever (38.1˚C) and AST and ALT elevation. None of these adverse events were related to the use of anti-adhesive agents. Other adverse events included neck pain, hypothyroidism, hypoparathyroidism, and hypocalcemia. All these adverse events were not related to the usage of the anti-adhesive agents. Table 7 presents the summary of adverse events by organ system classification.

## Discussion

This study was conducted through a randomized, controlled, double-blinded, multicenter, non-inferiority trial in order to evaluate the efficacy and safety of MegaShield®, a newly developed temperature-sensitive anti-adhesive agent. The efficacy and safety of ADM in prevention of adhesion formation after thyroid surgery was demonstrated by Kang et al., using ADM membrane (MegaDerm; L&C BIO, Seongnam, Korea). Although the study reported favorable results, the application of membrane-type anti-adhesive agent was considered more difficult than that of sol or gel forms, and adhesion formation at the suture site was of concern. Hence, we developed MegaShield®, a combination of temperature-sensitive sol-to-gel barrier and physical barrier with micronized ADM. In the control group, a commercially available temperature-sensitive anti-adhesive agent, Guardix-SG®, which was evaluated to be effective and safe in various surgical fields [28–30], but did not contain ADM was used.

**Table 3. Demographic and baseline characteristics (ITT set).**

| | MegaShield (Test device) | Guardix-SG (Control device) |
|---|---|---|
| | (N = 62) | (N = 65) |
| **Investigation center (n)** | | |
| Severance | 35 | 35 |
| KangBuk Samsung | 19 | 22 |
| Seoul St. Mary's Hospital | 8 | 8 |
| **Age (years)** | | |
| Mean | 48.1 | 49.2 |
| SD | 11.2 | 10.8 |
| Median | 50.5 | 50.0 |
| Range | (24.0–68.0) | (24.0–67.0) |
| **Weight (kg)** | | |
| Mean | 66.7 | 65.7 |
| SD | 13.9 | 14.5 |
| Median | 64.9 | 64.0 |
| Range | (40.0–100.3) | (44.0–118.0) |
| **Height (cm)** | | |
| Mean | 163.0 | 162.4 |
| SD | 7.9 | 8.5 |
| Median | 161.0 | 161.0 |
| Range | (147.0–184.0) | (145.0–186.0) |
| **BMI (kg/m$^2$)** | | |
| Mean | 25.0 | 24.7 |
| SD | 4.0 | 3.9 |
| Median | 24.4 | 24.3 |
| Range | (15.6–34.3) | (18.4–36.4) |
| **SBP (mmHg)** | | |
| Mean | 125.5 | 126.5 |
| SD | 10.8 | 13.3 |
| Median | 126.0 | 130.0 |
| Range | (97.0–146.0) | (99.0–150.0) |
| **DBP (mmHg)** | | |
| Mean | 77.3 | 78.8 |
| SD | 8.0 | 9.4 |
| Median | 79.0 | 80.0 |
| Range | (54.0–100.0) | (59.0–97.0) |
| **PR (beats/min)** | | |
| Mean | 81.5 | 78.9 |
| SD | 10.2 | 10.7 |
| Median | 82.0 | 77.0 |
| Range | (54.0–105.0) | (56.0–102.0) |
| **Sex (n [%])** | | |
| Male | 14 (22.6) | 16 (24.6) |
| Female | 48 (77.4) | 49 (75.4) |
| **Smoking (n [%])** | | |
| Yes | 1 (1.6) | 1 (1.5) |
| No | 61 (98.4) | 64 (98.5) |
| **Drinking (n [%])** | | |

(*Continued*)

**Table 3.** (Continued)

| | MegaShield (Test device) | Guardix-SG (Control device) |
|---|---|---|
| | (N = 62) | (N = 65) |
| Yes | 10 (16.1) | 7 (10.8) |
| No | 52 (83.9) | 58 (89.2) |
| **Any medical history (n [%])** | | |
| Yes | 46 (74.2) | 45 (69.2) |
| No | 16 (25.8) | 20 (30.8) |

In this study, the primary outcome was esophageal movement using marshmallow, which was objective, and the secondary outcome was the adhesion score, which was subjective. Although it is difficult to calculate the amount of adhesion, the phenotype of adhesion which is represented by dysfunction or certain symptoms of the patients can be evaluated by several measurements, and esophageal movement using marshmallow can be one of the choices. This is known to be useful as one of the good functional methods for detecting esophageal dysmobility and swallowing difficulty, which is triggered by the abnormal pharyngeal and upper esophageal muscle movement induced by post-thyroidectomy adhesion [31, 32].

The MegaShield® and Guardix-SG® groups did not show statistically significant difference in each criterion. In non-inferiority study, both ITT and PP group results are important [33], and we confirmed that they yield the same results. In addition, the trend of the adhesion score was noticeable. The adhesion score is a parameter to evaluate discomforts from postoperative adhesion such as swallowing difficulty, skin wrinkle, and scar formation, in which a higher score implies a more severe adhesion. In both MegaShield® and Guardix-SG® groups, the adhesion scores at visit 3 scored the highest and showed a decrease at visit 4. According to previous studies [34, 35], although scar remodeling process persists for more than two months,

**Table 4. Summary of marshmallow esophagography results at six weeks after thyroid gland surgery, visit 4 (day 42±7).**

| | mITT set | | | PP set | | |
|---|---|---|---|---|---|---|
| | MegaShield (Test device) (N = 62) | Guardix-SG (Control device) (N = 65) | Mean Difference (97.5% one-sided CI) | MegaShield (Test device) (N = 61) | Guardix-SG (Control device) (N = 65) | Mean Difference (97.5% one-sided CI) |
| **Marshmallow esophagography** | | | | | | |
| Mean | 2.8 | 2.6 | **0.16 (-0.08–Inf.)** | 2.8 | 2.6 | **0.17 (-0.07–Inf.)** |
| SD | 0.6 | 0.8 | | 0.6 | 0.8 | |
| SE | 0.1 | 0.1 | | 0.1 | 0.1 | |
| Median | 3.0 | 3.0 | | 3.0 | 3.0 | |
| IQR | 0.0 | 0.0 | | 0.0 | 0.0 | |
| Range | (0.0–3.0) | (0.0–3.0) | | (0.0–3.0) | (0.0–3.0) | |
| Score frequency | | | | | | |
| Normal (3) | 51 (82.3) | 49 (75.4) | | 51 (83.6) | 49 (75.4) | |
| Mild (2) | 9 (14.5) | 11 (16.9) | | 8 (13.1) | 11 (16.9) | |
| Moderate (1) | 1 (1.6) | 1 (1.5) | | 1 (1.6) | 1 (1.5) | |
| Severe (0) | 1 (1.6) | 4 (6.2) | | 1 (1.6) | 4 (6.2) | |

SD: standard deviation; SE: standard error; range: (Minimum–Maximum).

CI: confidence interval for the mean difference (test device—control device) based on unpaired t-test.

Predefined non-inferiority margin (-δ): -0.1.

**Table 5. Summary of total adhesion scores by postoperative time.**

| Postoperative time | MegaShield (Test device) | | | | | Guardix-SG (Control device) | | | | | [b]p-value | [c]p-value based on WMWodds |
|---|---|---|---|---|---|---|---|---|---|---|---|---|
| | N | Mean | SD | Median | [a]p-value | N | Mean | SD | Median | [a]p-value | | |
| mITT set | | | | | | | | | | | | |
| Screening | 62 | 8.82 | 4.02 | 6.00 | | 65 | 9.02 | 4.40 | 6.00 | | 0.816 | 0.7955 |
| Visit 3 (Day 7±3) | 62 | 13.10 | 7.82 | 8.00 | | 65 | 13.74 | 7.84 | 9.00 | | 0.135 | 0.6423 |
| Change from screening | 62 | 4.27 | 4.48 | 2.00 | < 0.001 | 65 | 4.72 | 4.46 | 3.00 | < 0.001 | 0.166 | 0.5698 |
| Visit 4 (Day 42±7) | 62 | 11.47 | 5.30 | 9.00 | | 65 | 11.89 | 5.38 | 9.00 | | 0.612 | 0.6523 |
| Change from screening | 62 | 2.65 | 3.14 | 2.00 | < 0.001 | 65 | 2.88 | 3.42 | 2.00 | < 0.001 | 0.704 | 0.6893 |
| PP set | | | | | | | | | | | | |
| Screening | 61 | 8.85 | 4.05 | 6.00 | | 65 | 9.02 | 4.40 | 6.00 | | 0.784 | 0.8278 |
| Visit 3 (Day 7±3) | 61 | 13.20 | 7.84 | 8.00 | | 65 | 13.74 | 7.84 | 9.00 | | 0.165 | 0.6970 |
| Change from screening | 61 | 4.34 | 4.48 | 2.00 | < 0.001 | 65 | 4.72 | 4.46 | 3.00 | < 0.001 | 0.212 | 0.6327 |
| Visit 4 (Day 42±7) | 61 | 11.54 | 5.32 | 9.00 | | 65 | 11.89 | 5.38 | 9.00 | | 0.685 | 0.7111 |
| Change from screening | 61 | 2.69 | 3.14 | 2.00 | < 0.001 | 65 | 2.88 | 3.42 | 2.00 | < 0.001 | 0.801 | 0.7464 |

SD: standard deviation.

[a] p-value for the comparison of change from screening in each device group by Wilcoxon signed-rank test at each time point.

[b] p-value for the comparison between two device groups by Wilcoxon rank sum test at each time point.

[c] p-value for testing WMWodds equal 1(Divine et al., 2018) at each time point.

**Table 6. Summary of adverse events (ITT set).**

| | MegaShield (Test device) (N = 70) | Guardix-SG (Control device) (N = 70) | Chi-square (P-value) |
|---|---|---|---|
| **Patients with any adverse event** | 35 (50.0) [82] | 38 (54.3) [91] | **0.11 (0.735)** |
| **Patients with serious adverse events** | 2 (2.9) [4] | - | **0.02 (0.181)**[a] |
| **Severity** | | | |
| Mild | 35 (50.0) [80] | 37 (52.9) [90] | |
| Moderate | 1 (1.4) [2] | 1 (1.4) [1] | |
| Severe | - | - | |
| **ID relationship** | | | |
| Definitely related | - | - | |
| Probably related | - | - | |
| Possibly related | - | - | |
| Probably not related | - | 1 (1.4) [2] | |
| Not related | 35 (50.0) [82] | 38 (54.3) [89] | |
| Unknown | - | - | |
| **Outcome** | | | |
| Recovered/Resolved | 35 (50.0) [74] | 37 (52.9) [79] | |
| Recovering/Resolving | - | 1 (1.4) [1] | |
| Not recovered/Not resolved | 8 (11.4) [8] | 10 (14.3) [11] | |
| Recovered/Resolved with sequelae | - | - | |
| Fetal | - | - | |
| Unknown | - | - | |

N (%) [] = number of patients reporting at least one adverse event in that category (percentage of patients among total N) [number of adverse-event occurrences].

**Table 7. Summary of adverse events by organ system classification (ITT set).**

| System Organ Class | MegaShield (Test device) | Guardix-SG (Control device) |
|---|---|---|
| | (N = 70) | (N = 70) |
| **Total** | **35 (50.0) [82]** | **38 (54.3) [91]** |
| **Musculoskeletal and connective tissue disorders** | **35 (50.0) [35]** | **35 (50.0) [36]** |
| Neck pain | 35 (50.0) [35] | 34 (48.6) [35] |
| Musculoskeletal pain | - | 1 (1.4) [1] |
| **Endocrine disorders** | **12 (17.1) [15]** | **13 (18.6) [16]** |
| Hypothyroidism | 8 (11.4) [8] | 9 (12.9) [9] |
| Hypoparathyroidism | 6 (8.6) [6] | 7 (10.0) [7] |
| Hyperthyroidism | 1 (1.4) [1] | - |
| **Metabolism and nutrition disorders** | **11 (15.7) [11]** | **12 (17.1) [12]** |
| Hypocalcemia | 11 (15.7) [11] | 12 (17.1) [12] |
| **Gastrointestinal disorders** | **3 (4.3) [4]** | **9 (12.9) [10]** |
| Nausea | 3 (4.3) [3] | 5 (7.1) [5] |
| Constipation | 1 (1.4) [1] | 4 (5.7) [4] |
| Dysphagia | - | 1 (1.4) [1] |
| **Investigations** | **3 (4.3) [6]** | **5 (7.1) [9]** |
| Aspartate aminotransferase increased | 2 (2.9) [3] | 5 (7.1) [5] |
| Alanine aminotransferase increased | 2 (2.9) [2] | 4 (5.7) [4] |
| Blood creatine increased | 1 (1.4) [1] | - |
| **Injury, poisoning and procedural complications** | **6 (8.6) [6]** | **4 (5.7) [4]** |
| Post procedural swelling | 4 (5.7) [4] | 3 (4.3) [3] |
| Post procedural edema | 1 (1.4) [1] | 1 (1.4) [1] |
| Post procedural hematoma | 1 (1.4) [1] | - |
| **General disorders and administration site conditions** | **4 (5.7) [4]** | **1 (1.4) [1]** |
| Pyrexia | 4 (5.7) [4] | 1 (1.4) [1] |
| **Infections and infestations** | **1 (1.4) [1]** | **1 (1.4) [1]** |
| Tonsillitis | 1 (1.4) [1] | 1 (1.4) [1] |
| **Nervous system disorders** | - | **1 (1.4) [1]** |
| Hypoesthesia | - | 1 (1.4) [1] |
| **Vascular disorders** | - | **1 (1.4) [1]** |
| Hypertension | - | 1 (1.4) [1] |

N (%) [] = number of patients reporting at least one adverse event in that category (percentage of patients among total N) [number of adverse-event occurrences].

most of the significant scar changes can develop within two to three months after the surgery. However, preventive management for postoperative adhesion such as pulse dye laser or application of a silicone gel sheet were available after the 6-week period, and it was ethically difficult to deny the patients of the chance of additional treatments.

In our previous study [19], a histological analysis revealed that the deposition of collagen and elastin fibers increased at six months compared to that at one month after ADM implantation. Moreover, microvessel density increased at three and six months compared to that at one month after ADM implantation. These results suggest that slower and controlled extracellular matrix (ECM) remodeling lead to long-term structural integrity and increased durability. Although this trial focused on the acute phase of scar changes after open thyroidectomy, these findings suggest that the efficacy of MegaShield® which includes micronized ADM can be increased during long-term follow up.

In the safety assessment, there were no device-related complications. There were two patients who had serious adverse events in the MegaShield® group. Both of these patients had high fever (>38˚C) without symptoms, in which one patient showed moderate increase in AST/ALT levels. Postoperative fever or AST/ALT elevation can be induced by analgesic agents used during the surgery, and we could not find direct connection between the application of the device and the adverse events. In the Guardix-SG® group, five cases of AST elevation and four cases of ALT elevation were counted as adverse events, in which none of them were device-related.

This study presents several limitations. First, there is no overall agreement on the criteria used for the adhesion score. The grading system we used is subjective and may not reflect the precise degree of adhesion. However, we also evaluated the objective assessment with the marshmallow esophagogram which can compensate the subjective grading system. Moreover, subjective symptoms of the patient can be considered as an important parameter as we can assess the effect of adhesion by the symptoms. Second, this study might require a subgroup analysis to determine if there is a difference in the results between the institutions; however, the number of cases was not enough for subgroup analysis. Third, we used Wilcoxon rank-sum test for the statistical analysis of this study, based on its widespread use to analyze non-normally distributed data, with the Shapiro-Wilk test. Although it is not entirely suitable as a median test, we applied the Wilcoxon rank-sum test to demonstrate if the two populations are equally distributed or not [36]. Fourth, this trial focused on the acute phase of scar changes after open thyroidectomy due to ethical issues. Although we did not restrict additional anti-adhesive treatment after the six-week period, an observational study for the two groups after the follow-up period may be conducted in future.

## Conclusion

This randomized, controlled, double-blind, multicenter trial demonstrated that the efficacy and safety of MegaShield® was not inferior compared to that of Guardix-SG®. MegaShield® demonstrated the potential of ADM as a potential future anti-adhesive agent.

## Supporting information

**S1 Checklist. CONSORT 2010 checklist of information to include when reporting a randomised trial\*.**
(DOC)

**S1 Table. Total setting of the enrolled patients.**
(DOCX)

**S1 File.**
(PDF)

**S2 File.**
(PDF)

**S1 Data.**
(XLSX)

## Author Contributions

**Conceptualization:** Kee-Hyun Nam.

**Data curation:** Jin Kyong Kim, Sang-Wook Kang, Jong Ju Jeong, Kee-Hyun Nam, Sung-Rae Cho, Seongmoon Jo, Eun Young Kim, Ji-Sup Yun, Hee Jin Park, Mi Sung Kim, Kwangsoon Kim, Sohee Lee, Ja Seong Bae, So Yeon Jun, Jihye Park, Jeong Soo Kim.

**Formal analysis:** Jin Kyong Kim, Sang-Wook Kang, Kee-Hyun Nam.

**Funding acquisition:** Kee-Hyun Nam.

**Investigation:** Cho Rok Lee, Kee-Hyun Nam.

**Methodology:** Kee-Hyun Nam.

**Project administration:** Kee-Hyun Nam, Ji-Sup Yun, Ja Seong Bae.

**Resources:** Kee-Hyun Nam.

**Software:** Jin Kyong Kim, Cho Rok Lee, Sang-Wook Kang, Jong Ju Jeong, Kee-Hyun Nam, Sung-Rae Cho, Seongmoon Jo, Eun Young Kim, Ji-Sup Yun, Hee Jin Park, Mi Sung Kim, Kwangsoon Kim, Sohee Lee, Ja Seong Bae, So Yeon Jun, Jihye Park, Jeong Soo Kim.

**Supervision:** Jong Ju Jeong, Kee-Hyun Nam.

**Validation:** Cho Rok Lee, Kee-Hyun Nam.

**Visualization:** Jin Kyong Kim, Kee-Hyun Nam.

**Writing – original draft:** Jin Kyong Kim.

**Writing – review & editing:** Jong Ju Jeong, Kee-Hyun Nam.

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
