## [Decision Letter · Decision Letter 0]

21 Sep 2021

PONE-D-21-04211Efficacy and Safety of Temperature-sensitive Acellular Dermal Matrix for Prevention of Postoperative Adhesion after Thyroidectomy: A Randomized, Multicenter, Double-blind studyPLOS ONE

Dear Dr. Nam,

Thank you for submitting your manuscript to PLOS ONE. After careful consideration, we feel that it has merit but does not fully meet PLOS ONE’s publication criteria as it currently stands. Therefore, we invite you to submit a revised version of the manuscript that addresses the points raised during the review process.

 The reviewers have identified a number of concerns that need to be carefully addressed in a revision to your manuscript.

We look forward to receiving your revised manuscript.

Kind regards,

Jamie Males

Staff Editor

PLOS ONE

Journal Requirements:

3. We note from your protocol that this study was sponsored by L&C Bio. Please review the PLOS competing interests and financial declaration policies at https://journals.plos.org/plosone/s/competing-interests and https://journals.plos.org/plosone/s/disclosure-of-funding-sources, and update your financial declaration and competing interests statements to declare all financial support provided for this study.

Reviewers' comments:

Reviewer's Responses to Questions

**Comments to the Author**

1. Is the manuscript technically sound, and do the data support the conclusions?

Reviewer #1: Partly

Reviewer #2: Yes

2. Has the statistical analysis been performed appropriately and rigorously? 

Reviewer #1: Yes

Reviewer #2: Yes

3. Have the authors made all data underlying the findings in their manuscript fully available?

Reviewer #1: No

Reviewer #2: Yes

4. Is the manuscript presented in an intelligible fashion and written in standard English?

Reviewer #1: Yes

Reviewer #2: Yes

5. Review Comments to the Author

Reviewer #1: The authors reported the results of a randomized, multicenter, double-blind

Non-inferiority study to investigate the efficacy, determined by the extent of adhesion as assessed using Marshmallow esophagography 6 weeks after the surgery, of MegaShield ® compared with Guardix-SG ® in patients undergoing bilateral total thyroidectomy. They concluded, “The efficacy and safety of MegaShield ® was not inferior compared to that of Guardix-SG ®”.

In general I think the paper describes a well planned study, but the description of the result need to address the planning details much more specific. In the following. I will start with some more general comments:

• As the study aims to prove a non-inferiority hypothesis the appropriate CONSORT Checklist should be applied and the structure of the paper should follow the section/topics of the guideline. That helps by identification of the important information of this study type. Among other, the study typ as non-inferiority study should be included in the title. Please note also that ervey point in the guideline should be addressed, e.g. 3b should be commented as: No changes to methods after trial commencement are implemented. In particular this information is important as the trial register does not allow to follow the history of changes. See: Piaggio G (2012), PMID: 23268518.

• There is no appended individual patient level data file, so that reproducibility is limited.

• As most of the details, e.g. the non-inferiorty margin including justification, these information is included only on the side so that study objective is unclear most the time. This would be solved, if the CONSORT recommendation is followed strictly.

Detailed comments:

L83ff: Include the hypothesis and endpoint in the definition of the aim.

L105: Block size is missing in the description of PBR.

L105f: It is unclear whether the randomization is stratified by center.

L109: It is unclear how blinding is implemented. Please give details. I assume that treatments could not be blinded. Give details for blinding of clinical assessors.

L132: As biostatistican it is not clear to me, whether the rimary outcome in the manuskrip “extent of adhesion as assessed using Marshmallow esophagography” equals “esophageal movement using Marshmallow” or “” which is given in the protocol.

L161: Statistical tests on baseline characterisics is not. Useful in randomized trials. Delete column 4. Give numbers recruited patients in groups Per “center”.

L172f: Give more details for sample size calculation. I guess your argument is based on the z-statistic as in the protocol. Describe your arguments for the non-inferiority margin. Give arguments for the dropout rate, e.g. by referring to the literature. The ICH E9 stated to do the ITT and PP and they should yield the same results. SO you may proceed similarly by giving the two onesided 95% Cis. Actually, the method does not result in a conservative test decision and the method of Sanchez (Statistics in Medicine, 2006) may be a useful as an alternative. You just have to argue.

L183: I assume you mention statistical tests on the secondary endpoint etc. There are good arguments to avoid pretest for normality – so may be you delete that text. Second, the tests could not be associated to the parameters, so it is best to mention the test in the result section at places where it is used. Give the software used for statistical computation.

L187f: Describe the setting first. Report on enrollment time, patients per center and in particular deviations in the randomization process.

L193: I see the exclusion of 11 patients critical with regard to the ITT principle. As there are longitudinal measurements other strategies to handle missing observations may be more appropriate. In particular with regard to figure 1, where only 1 patient is withdrawn in each group.

L209f: Use the analysis population mentioned above. Skip the p-values.

- Give p-values with at least three decimal digits!

L226: Please improve the layout of the table5, so that no line break occurs in numbers. As the data are compared by Wilcoxon test, the appropriate statistic based on provision Probability should be stated (WMWodds, see Divine, 2018). On the other hand, the longitudinal measurements may be analysed by Mixed effects models, rather than splitting over time.

L288: Limitation should address and discuss potential sources of bias in the trial and success of implemented measures against bias.

L299: This is rather a result than a clinical conclusion. What does the result mean for practice?

Reviewer #2: Well designed study.

I would have liked a more detailed explanation of the marshmallow oesophagography technique. The transit time is used as a surrogate marker for adhesions. I would have liked to see some evidence or justification that an increased transit time correlates with the degree of adhesions.

The subjective scoring system to assess adhesions could be improved

6. PLOS authors have the option to publish the peer review history of their article (what does this mean?). If published, this will include your full peer review and any attached files.

Reviewer #1: No

Reviewer #2: No

---

## [Author Response · Author response to Decision Letter 0]

3 Nov 2021

November 4, 2021

Emily Chenette

Editor-in-Chief

PLOS ONE

Dear Editor:

Thank you for giving us the opportunity to submit a revised draft of our manuscript titled “Efficacy and Safety of Temperature-sensitive Acellular Dermal Matrix in Prevention of Postoperative Adhesion after Thyroidectomy: A Randomized, Multicenter, Double-blind, Non-inferiority study”

to PLOS ONE. We appreciate the time and effort that you and the reviewers have dedicated to providing valuable feedback regarding our manuscript. We are grateful to the reviewers for their insightful comments on our paper. We have been able to incorporate changes to reflect most of the suggestions provided by the reviewers. The changes made within the manuscript have been highlighted to aid navigation. 

Here are our point-by-point responses to the reviewers’ comments and concerns. 

Comments from Reviewer 1

Comment 1: As the study aims to prove a non-inferiority hypothesis the appropriate CONSORT Checklist should be applied and the structure of the paper should follow the section/topics of the guideline. That helps by identification of the important information of this study type. Among other, the study type as non-inferiority study should be included in the title. Please note also that ervey point in the guideline should be addressed, e.g. 3b should be commented as: No changes to methods after trial commencement are implemented. In particular this information is important as the trial register does not allow to follow the history of changes. See: Piaggio G (2012), PMID: 23268518.

Response: Thank you for your valuable comment. In this revision, we did our best to comply with the CONSORT guidelines. We have also commented, “No changes to methods after trial commencement are implemented” in the methods section (page 4, paragraph 3, line 100). 

Comment 2

There is no appended individual patient level data file, so that reproducibility is limited.

Response: It is not clear as to what the individual patient level data imply, as the PLOS ONE guidelines do not specifically obligate it. However, we have provided the score frequency in Table 4, which made it easier to analyze the primary outcome and arrive at the same conclusion. 

Comment 3

As most of the details, e.g. the non-inferiorty margin including justification, these information is included only on the side so that study objective is unclear most the time. This would be solved, if the CONSORT recommendation is followed strictly

Response: Thank you for your comment which made the structure of this manuscript more systematic. As we have commented above, we have tried to follow the CONSORT recommendation strictly in the revised manuscript.

Comment 4

L83ff: Include the hypothesis and endpoint in the definition of the aim. 

Response: Thank you for your comment. Accordingly, we have described the hypothesis and outcomes (page 4, paragraph 2, line 86–88).

Comment 5

L105: Block size is missing in the description of PBR.

Response: Thank you for your comment. Accordingly, we have added the information on the block size. (page 5, paragraph 2, line 111).

Comment 6

L105f: It is unclear whether the randomization is stratified by center.

Response: According to your comment, we have clearly stated that the randomization is stratified by center. (page 5, paragraph 2, line 111).

Comment 7

L109: It is unclear how blinding is implemented. Please give details. I assume that treatments could not be blinded. Give details for blinding of clinical assessors.

Response: Thank you for your valuable comment. We agree that the implementation of blinding can be explained more clearly based on the study protocol. We have added this information in the manuscript. (page 5, paragraph 2, line 114–121).

Comment 8

L132: As biostatistican it is not clear to me, whether the primary outcome in the manuskrip “extent of adhesion as assessed using Marshmallow esophagography” equals “esophageal movement using Marshmallow” or “” which is given in the protocol. 

Response: Thank you for your insightful comment. We agree that the similarity of the expressions should be resolved. We think that the “extent of adhesion as assessed using marshmallow esophagography” is what we wanted to evaluate with “esophageal movement using marshmallow”. We have changed the expression in the manuscript to “esophageal movement using marshmallow”, following the expression in the guideline. To make this more logical, we have explained it in the discussion section. (page 17, paragraph 2, line 298–304).

Comment 9

L161: Statistical tests on baseline characterisics is not. Useful in randomized trials. Delete column 4. Give numbers recruited patients in groups Per “center”.

Response: Thank you for your suggestion. As per your comment, we have modified Table 1. 

Comment 10

L172f: Give more details for sample size calculation. I guess your argument is based on the z-statistic as in the protocol. Describe your arguments for the non-inferiority margin. Give arguments for the dropout rate, e.g. by referring to the literature. The ICH E9 stated to do the ITT and PP and they should yield the same results. SO you may proceed similarly by giving the two one sided 95% Cis. Actually, the method does not result in a conservative test decision and the method of Sanchez (Statistics in Medicine, 2006) may be a useful as an alternative. You just have to argue.

Response: Thank you for your suggestion. I agree that a detailed explanation of the protocol should be presented in the manuscript. We have described it in the manuscript with the arguments for non-inferiority margin (page 8, paragraph 1, lines 168–174). Regarding the ICH E9, we have commented that we obtained the same results with both ITT and PP groups with the reference that you had recommended (page 17, paragraph 3, lines 305–307).

Comment 11

L183: I assume you mention statistical tests on the secondary endpoint etc. There are good arguments to avoid pretest for normality – so maybe you delete that text. Second, the tests could not be associated to the parameters, so it is best to mention the test in the result section at places where it is used. Give the software used for statistical computation

Response: I appreciate your professional comment. Based on your comment, we have deleted the relevant sentence. We included the details of the software used for statistical computation (SAS ver. 9.4; SAS Institute, NC, USA) at the beginning of the paragraph (page 8, paragraph 1, line 183-184). Further, if possible, we would like to add footnotes to the tables pertaining to the correlated statistical test to add clarity to the manuscript.

Comment 12

L187f: Describe the setting first. Report on enrollment time, patients per center, and in particular deviations in the randomization process.

Response: I appreciate your comment. We have described the setting including enrollment time and patients per center. We have also commented that no deviations occurred during the randomization process. (page 8, paragraph 2, lines 188–189 and page 9, paragraph 1, lines 191–193).

Comment 13

L193: I see the exclusion of 11 patients critical with regard to the ITT principle. As there are longitudinal measurements other strategies to handle missing observations may be more appropriate. In particular with regard to figure 1, where only 1 patient is withdrawn in each group. 

Response: We admit that Figure 1 may be appear confusing to the readers. Therefore, we have revised it to explain the whole patient setting. Further, the data of the patients who were included in ITT but excluded in mITT were not formally approved at the time of the first postoperative assessment (visit 3). Alternatively, there were no missing observations to substitute in mITT group.

Comment 14

L209f: Use the analysis population mentioned above. Skip the p-values. Give p-values with at least three decimal digits!

Response: As we have commented previously, it seems that the analysis of mITT group does not contain statistical problems. We have described this in the discussion section, which also corresponds to the answer to your comment 10. 

Comment 15

L226: Please improve the layout of the table 5, so that no line break occurs in numbers. As the data are compared by Wilcoxon test, the appropriate statistic based on provision Probability should be stated (WMWodds, see Divine, 2018). On the other hand, the longitudinal measurements may be analysed by Mixed effects models, rather than splitting over time.

Response: We thoroughly read the article written by Divine et al. Indeed, the Wilcoxon test has a limitation as a test of medians. However, this study aimed to evaluate the difference between two groups rather than perform a median test. As we had planned to use the Wilcoxon-Mann-Whitney test for statistical analysis in this study, based on its widespread usage for analysis of non-normally distributed data, we would like to retain the statistical results. We have described this as a limitation in the discussion section (page 18, paragraph 4, lines 334– page 19, paragraph 1, line 337). 

Comment 16

L288: Limitation should address and discuss potential sources of bias in the trial and success of implemented measures against bias

Response: We agree that we must add a comment pertaining to the potential source of bias in this study. However, as evident from the study guideline (11.3.1), we attempted to minimize the statistical biases with double-blinding and randomization generated using SAS by a statistical expert who was not directly related to the institution. For blinding, before the study was terminated, only a minimum number of personnel who are not directly related to the institution were able to access the randomization table and code. Regarding financial bias, this study was stated to be sponsored by L&C bio in the protocol; however, the study grant was finally sponsored by the Ministry of Health & Welfare as mentioned in the financial disclosure.

Comment 17

L299: This is rather a result than a clinical conclusion. What does the result mean for practice? 

‘Response: We appreciate your comment. We have added a sentence to the conclusion to show the clinical usefulness of this study (page 19, paragraph 2, line 342–344). 

Comments from Reviewer 2

Comment 1

Well designed study.

I would have liked a more detailed explanation of the marshmallow oesophagography technique. The transit time is used as a surrogate marker for adhesions. I would have liked to see some evidence or justification that an increased transit time correlates with the degree of adhesions.

The subjective scoring system to assess adhesions could be improved.

Response: We appreciate your comment. We agree that the association between the extent of adhesion and the transit time on marshmallow esophagography can be explained more thoroughly. Based on your comment, we have added more references and relevant explanations. (page 17, paragraph 2, lines 298–304).

In addition to the above comments, all spelling and grammatical errors have been corrected. 

We look forward to hearing from you in due course regarding our submission and will be glad to respond to any further questions and comments that you may have. 

Kee-Hyun Nam

Address: Department of Surgery, Yonsei University College of Medicine, 

50-1, Yonsei-ro, Seodaemun-gu, Seoul, Republic of Korea

Phone: +82-2-2228-2100

Email: khnam@yuhs.ac

---

## [Decision Letter · Decision Letter 1]

17 Dec 2021

PONE-D-21-04211R1Efficacy and Safety of Temperature-sensitive Acellular Dermal Matrix in Prevention of Postoperative Adhesion after Thyroidectomy: A Randomized, Multicenter, Double-blind, Non-inferiority studyPLOS ONE

Dear Dr. Nam,

Thank you for submitting your manuscript to PLOS ONE. After careful consideration, we feel that it has merit but does not fully meet PLOS ONE’s publication criteria as it currently stands. Therefore, we invite you to submit a revised version of the manuscript that addresses the points raised during the review process.

The manuscript has been evaluated by two reviewers, and their comments are available below.<o:p></o:p>

One of the reviewers has raised a minor concern. They feel the manuscript results section should be edited to better present the statistical analysis. <o:p></o:p>

Could you please carefully revise the manuscript to address all comments raised?

We look forward to receiving your revised manuscript.

Kind regards,

Thomas Phillips, PhD

Staff Editor

PLOS ONE

Journal Requirements:

Reviewers' comments:

Reviewer's Responses to Questions

**Comments to the Author**

1. If the authors have adequately addressed your comments raised in a previous round of review and you feel that this manuscript is now acceptable for publication, you may indicate that here to bypass the “Comments to the Author” section, enter your conflict of interest statement in the “Confidential to Editor” section, and submit your "Accept" recommendation.

Reviewer #1: (No Response)

Reviewer #2: All comments have been addressed

2. Is the manuscript technically sound, and do the data support the conclusions?

Reviewer #1: Yes

Reviewer #2: Yes

3. Has the statistical analysis been performed appropriately and rigorously? 

Reviewer #1: No

Reviewer #2: Yes

4. Have the authors made all data underlying the findings in their manuscript fully available?

Reviewer #1: No

Reviewer #2: Yes

5. Is the manuscript presented in an intelligible fashion and written in standard English?

Reviewer #1: Yes

Reviewer #2: Yes

6. Review Comments to the Author

Reviewer #1: Thank you for considering my comments. There is only one point to be addressed in the final review. The presentation of the results of the Wilcoxon test needs providing the appropriate statistic based on provision Probability , i.e. WMWodds (see Divine, 2018).

Reviewer #2: The previous comment of the marshmallow esophagography transit time had been addressed and this section appears clearer in the paper

7. PLOS authors have the option to publish the peer review history of their article (what does this mean?). If published, this will include your full peer review and any attached files.

Reviewer #1: No

Reviewer #2: No

---

## [Author Response · Author response to Decision Letter 1]

2 Jun 2022

Journal Requirements

Response: As per your comment, we have changed or deleted several references. The four references listed below were not retracted articles, but were published in Korean journals, which means that they were not internationally acceptable. We apologize for our disregard for proper referencing. 

Reference 3 (Oh et al. Trends of anti-adhesion adjuvant-review) was deleted. The journal (biomaterials research) was originally a Korean journal and became an international journal after 2014. Unfortunately, the reference we had used was from 2013. We found that deleting this reference made no difference in the content, so we decided to delete this article from the reference list. 

Reference 12 (Lee et al. The efficacy and safety of HA/CMC anti-adhesion barrier solution with varying viscosities) was changed to reference 11, which is a reference from an international journal (Leach et al, Reduction of postsurgical adhesion formation in the rabbit uterine horn model with use of hyaluronate/carboxymethylcellulose gel. Fertility and sterility. 1998;69: 415-418).

Reference 26 and 27 (Song et al. Clinical value of marshmallow esophagography in detecting esophageal dysmotility and Park et al. Significance of functional esophagogram with a marshmallow bolus) were changed to reference 25, which is an internationally acceptable reference (Ott et al. Evaluation of the esophagus with a marshmallow bolus: Clarifying the cause of dysphagia. Gastrointestinal radiology. 1991;16: 1-4 

Comments from Reviewer 1

Comment 1: Thank you for considering my comments. There is only one point to be addressed in the final review. The presentation of the results of the Wilcoxon test needs providing the appropriate statistic based on provision Probability, i.e. WMWodds (see Divine, 2018).

Response: We thank the reviewer for this valuable comment regarding the appropriate reference for the latest analysis. Accordingly, the results are now presented in Table 5 (page 13). We used the SAS program provided in Divine et al., (2018). We hope that this revision is in accordance with the statistical standards of the journal.

---

## [Decision Letter · Decision Letter 2]

5 Aug 2022

Efficacy and Safety of Temperature-sensitive Acellular Dermal Matrix in Prevention of Postoperative Adhesion after Thyroidectomy: A Randomized, Multicenter, Double-blind, Non-inferiority study

PONE-D-21-04211R2

Dear Dr. Nam,

We’re pleased to inform you that your manuscript has been judged scientifically suitable for publication and will be formally accepted for publication once it meets all outstanding technical requirements.

Kind regards,

Miquel Vall-llosera Camps

Senior Editor

PLOS ONE

Reviewers' comments:

Reviewer's Responses to Questions

**Comments to the Author**

1. If the authors have adequately addressed your comments raised in a previous round of review and you feel that this manuscript is now acceptable for publication, you may indicate that here to bypass the “Comments to the Author” section, enter your conflict of interest statement in the “Confidential to Editor” section, and submit your "Accept" recommendation.

Reviewer #1: All comments have been addressed

Reviewer #2: All comments have been addressed

2. Is the manuscript technically sound, and do the data support the conclusions?

Reviewer #1: Yes

Reviewer #2: Yes

3. Has the statistical analysis been performed appropriately and rigorously? 

Reviewer #1: Yes

Reviewer #2: Yes

4. Have the authors made all data underlying the findings in their manuscript fully available?

Reviewer #1: Yes

Reviewer #2: Yes

5. Is the manuscript presented in an intelligible fashion and written in standard English?

Reviewer #1: Yes

Reviewer #2: Yes

6. Review Comments to the Author

Reviewer #1: Substitute "p-value for the median comparison between two device groups by WMWodds (Divine et al., 2018) at each time point" by

p-value for testing WMWodds equal 1 (Divine et al., 2018) at each time point

Reviewer #2: The revised manuscript has addressed all the comments raised previously. Well written and researched

7. PLOS authors have the option to publish the peer review history of their article (what does this mean?). If published, this will include your full peer review and any attached files.

Reviewer #1: No

Reviewer #2: No

---

## [Editor Report · Acceptance letter]

9 Sep 2022

PONE-D-21-04211R2 

Efficacy and Safety of Temperature-sensitive Acellular Dermal Matrix in Prevention of Postoperative Adhesion after Thyroidectomy: A Randomized, Multicenter, Double-blind, Non-inferiority study 

Dear Dr. Nam:

I'm pleased to inform you that your manuscript has been deemed suitable for publication in PLOS ONE. Congratulations! Your manuscript is now with our production department. 

Kind regards, 

on behalf of

Dr Thomas Phillips 

Staff Editor

PLOS ONE